**Data Availability Statement:** All relevant data are within the paper and its Supporting information files (file name: Anonymized Study Data).

# Health care seeking patterns of rifampicin-resistant tuberculosis patients in Harare, Zimbabwe: A prospective cohort study

Rebecca Tadokera[1], Stella Huo[2], Grant Theron[1], Collins Timire[3,4], Salome Manyau-Makumbirofa[5‡], John Z. Metcalfe[2‡]*

1 Division of Molecular Biology and Human Genetics, Faculty of Medicine and Health Sciences, NRF/DST Centre of Excellence for Biomedical Tuberculosis Research, South African Medical Research Council Centre for Tuberculosis Research, Stellenbosch University, Cape Town, South Africa, 2 Division of Pulmonary and Critical Care Medicine, San Francisco General Hospital, University of California, San Francisco, San Francisco, California, United States of America, 3 Ministry of Health and Child Care, National Tuberculosis Control Programme, Harare, Zimbabwe, 4 International Union Against Tuberculosis and Lung Disease Zimbabwe Office, Centre for Operational Research, Harare, Zimbabwe, 5 Department of Global Health and Development, Faculty of Public Health and Policy, London School of Hygiene and Tropical Medicine, London, United Kingdom

‡ These authors are joint senior authors on this work.
* John.Metcalfe@ucsf.edu

## Abstract

### Background

Delays in seeking and accessing treatment for rifampicin-resistant tuberculosis (RR-TB) and multi-drug resistant (MDR-TB) are major impediments to TB control in high-burden, resource-limited settings.

### Method

We prospectively determined health-seeking behavioural patterns and associations with treatment outcomes and costs among 68 RR-TB patients attending conveniently selected facilities in a decentralised system in Harare, Zimbabwe.

### Results

From initial symptoms to initiation of effective treatment, patients made a median number of three health care visits (IQR 2–4 visits) at a median cost of 13% (IQR 6–31%) of their total annual household income (mean cost, US$410). Cumulatively, RR-TB patients most frequently first visited private facilities, i.e., private pharmacies (30%) and other private health care providers (24%) combined. Median patient delay was 26 days (IQR 14–42 days); median health system delay was 97 days (IQR 30–215 days) and median total delay from symptom onset to initiation of effective treatment was 132 days (IQR 51–287 days). The majority of patients (88%) attributed initial delay in seeking care to "not feeling sick enough." Total delay, total cost and number of health care visits were not associated with treatment or clinical outcomes, though our study was not adequately powered for these determinations.

**Funding:** This work was supported in part by the National Institutes of Health (K23 AI094251 to J.Z. M.), the Robert Wood Johnson Foundation (Amos Medical Faculty Development Program Award to J. Z.M.). RT and CT were supported by the Fogarty International Center of the National Institutes of Health under Award Number D43 TW009539. The funders had no role in study design, data collection and analysis, decision to publish, or preparation of the manuscript. The work is solely the responsibility of the authors and does not necessarily represent the official views of the National Institutes of Health.

**Competing interests:** The authors have declared that no competing interests exist.

## Conclusions

Despite the public availability of rapid molecular TB tests, patients experienced significant delays and high costs in accessing RR-TB treatment. Active case finding, integration of private health care providers and enhanced service delivery may reduce treatment delay and TB associated costs.

## Introduction

Multi-drug resistant (MDR) tuberculosis (TB) remains a public health crisis and health security threat [1]. The social and economic burden associated with TB and MDR-TB treatment [2], compounded by the HIV epidemic, places a disproportionate burden of disease on Sub-Saharan African countries [3, 4]. TB diagnosis, treatment, and prevention are strongly associated with socio-economic and behavioural factors [5–7]. Operational challenges related to weak, poorly resourced health systems are a barrier to TB prevention and control activities. For MDR-TB, timing delays can significantly determine treatment outcomes and are likely to increase the infectiousness and disease transmission in a community [4]. The resultant delays to diagnosis and effective treatment related to health system weaknesses are further informed by patient choices and behaviour [8]. Ultimately, such delays increase community transmission and may worsen treatment outcomes [9].

Despite the availability of free TB treatment in the public sector and the goal of universal health coverage (UHC) [10], the economic consequences for families affected by TB are often severe [11]. These include direct medical costs, direct non-medical costs and income loss (resulting from indirect or opportunity costs) [12]. The World Health Organization (WHO) defines TB-associated health care expenditures above a certain proportion (typically 20%) of available annual income as "catastrophic", though the specific threshold is expected to vary by setting and circumstances [13, 14]. Resulting impoverishment may be high in settings such as Zimbabwe, where 85% of the total workforce is employed within the informal sector [15].

RR-TB has conventionally been used as a reliable proxy for MDR-TB, particularly in resource limited settings [16]. We conducted a prospective cohort study to understand health-care seeking behaviour, healthcare expenditures, and associations with treatment outcome (24-month survival) among RR-TB patients starting treatment in conveniently selected outpatient facilities in Harare, Zimbabwe.

## Materials and methods

### Study population

RR-TB patients initially seeking health care within the Harare Metropolitan area between November 2011 and November 2012 were recruited from eight conveniently selected health facilities. Detailed enrolment and diagnosis criteria have previously been described [17]. Briefly, persons who had a history of prior TB treatment or were suspected of having drug-resistant pulmonary TB were recruited into a prospective cohort study. Presumptive drug-resistant TB patients were identified as symptomatic patients presenting with a history of > = 1 month of prior TB treatment (relapse, treatment after loss to follow up or treatment failure), contact with a person with known or possible drug-resistant TB; or with a rifampicin-resistant result on Xpert MTB/RIF. RR-TB diagnosis was confirmed by molecular TB assay and/or phenotypic drug susceptibility testing. At the time our study was undertaken, a partially

decentralized MDR-TB/RR-TB treatment system was in place in Harare. Patients could only be diagnosed and recommended for treatment via a central MDR-TB clinic (i.e., at Wilkins Hospital) based in Harare, but MDR-TB/RR-TB treatment could be dispensed through peripheral, local health clinics. We estimated that approximately 60% of the population base was likely included in our sample [17].

## Data collection

Data were collected using a mixed methods approach via an open-ended healthcare-seeking behaviour questionnaire designed to capture both quantitative and qualitative responses through in-depth patient interviews. The open-ended questionnaire allowed respondents to provide detailed responses on their health-seeking pathways and reasons for delay in seeking care. A trained research nurse administered the questionnaire. Quantitative data included basic socio-demographic information. Health-seeking behaviour included type of facilities or health care provider from which treatment was first sought; delays in seeking health care; reasons for delays; and direct and indirect costs incurred. Additional variables included whether patients had been referred from outside Harare, HIV and antiretroviral treatment status, TB treatment history and TB symptoms. Final clinical outcome was defined as 24-month patient survival, ascertained through review of medical records and contact with next-of-kin. Data were electronically captured in Research Electronic Data Capture tools (REDCap) [18] hosted at the University of California, San Francisco.

## Definitions

Patient delay was defined as the time from onset of disease-associated symptoms to first health care visit. Health system delay was defined as the time from first medical facility visit to initiation of effective RR-TB treatment. Total delay was defined as the sum of patient and health system delays and included both diagnostic delay and repeat visits before initiation of treatment. Direct costs were those directly associated with health care such as diagnostic tests, doctors' fees or medication costs. Indirect costs included travel, lodging or food expenses indirectly associated with health care seeking. Total costs were defined as the sum of direct and indirect costs incurred by a patient accessing care. A polyclinic was defined as a patient's local government clinic (usually the first point of contact in this setting). Informal employment was defined as engagement in an economic activity that is not taxed or formally registered. Final clinical outcome was defined as 24-month patient survival.

## Statistical analysis

Statistical analysis was performed using Stata Version 13 (StataCorp, College Station, TX, USA) and R software (version 3.5.3). Descriptive statistics for numeric variables (such as age or income) were computed and are reported as median and interquartile range (IQR). Chi-square tests were used to compute associations between continuous and categorical variables. We used the Kruskal Wallis test to analyse the associations between health care facility first attended; delays (patient and total time delays) and cost (both first visit and total costs incurred) to patients. Mann Whitney tests were used to compute associations between delay and clinical outcomes. Results are reported as medians, IQRs or proportions.

Both simple and multiple linear regression analyses were used to assess the associations between delay and explanatory variables. Potential risk factors for delay and variables associated with delay in the bivariate analysis ($P \leq 0.2$) were included in a final multiple linear regression model. A p-value of less than 0.05 was considered statistically significant. R software (version 3.5.3) was used to compute the visual pathways of care. Thematic analysis was used to

identify and analyse patterns and trends within the qualitative data gathered from the open-ended questionnaires [19].

### Ethical approval

The Medical Research Council of Zimbabwe (MRCZ/A1552), and the Institutional Review Board of Biomedical Research and Training Institute, and Human Research Protection Programme, University of California, San Francisco (USCF) provided ethical approval (10–05115). All participants gave written informed consent before enrolling in the study.

## Results

### Study participants

Of 139 participants with RR-TB in the main study, 73 (53%) agreed to participate in the health-seeking behaviour (HSB) survey (Fig 1). A final sample of 68 participants with follow-up data was included in the final analysis: five participants were lost to follow up during the study period and were excluded from further analysis. More than 70% (48/68) of study participants were female, mainly from the economically active age group (median 34 years, IQR 29–42 years); with more than 73%, (50/68) educated up to secondary level (Table 1). Forty-five percent (30/67) of participants were employed in the informal sector; median monthly income was US$175 (IQR US$100-$300). Two-thirds of the sample (43/68) were HIV-positive, among whom 31 (73.8%) were on antiretroviral therapy. Overall, 37 (55.2%) of participants presented with new RR-TB, while other participants had experienced from one to three prior TB episodes (29.9% and 3.0% respectively).

### Time delays and costs associated with health care visits

Patients delayed seeking care by a median of 26 days (IQR 14–42 days) from onset of TB symptoms (Table 2). Median health system delay (i.e., time from first health care visit to initiation of effective RR-TB treatment) was 97 days (IQR 30–215 days). Health system delay was longest when the first point of contact was government polyclinics (median 150 days, IQR 18–300 days), followed by private clinics (median 102 days, IQR 30–154 days) and pharmacies (median 97 days, IQR 67–210 days). Median total delay (i.e., patient plus health system delay) was 132 days (IQR 51–287 days). Total delay was longest when a polyclinic was the first health system contact (median, 221 days, IQR 45–338 days). Costs associated with first visit were highest for private clinics (median US$30, IQR $14–45 days) and lowest for polyclinics (median $US0, IQR $0–4), across facilities visited (p<0.001). Cumulative costs were highest for participants who first visited a pharmacy (median US$45, IQR $19–80) and private clinics/hospital (median US$40, IQR $15–56), but least for those first visiting government polyclinics (median US$8, IQR $5–16). Overall, a median of 13% (IQR 6–31%) of total annual household income, (equivalent to a mean cost of US$410), was spent on seeking RR-TB health care. At least 30% (23/68) households incurred TB related costs which can be considered as catastrophic i.e., above 20% of annual household income.

### Reasons for delays in seeking care

Most study participants (n = 59/68, 88%) who delayed seeking care for their first health care visit reported not feeling sick enough to warrant a visit to the clinic or thinking that they had a minor cough, which could be treated using over-the-counter remedies. Participants who did not first seek care at primary care facilities (polyclinics) reported anticipating longer waiting

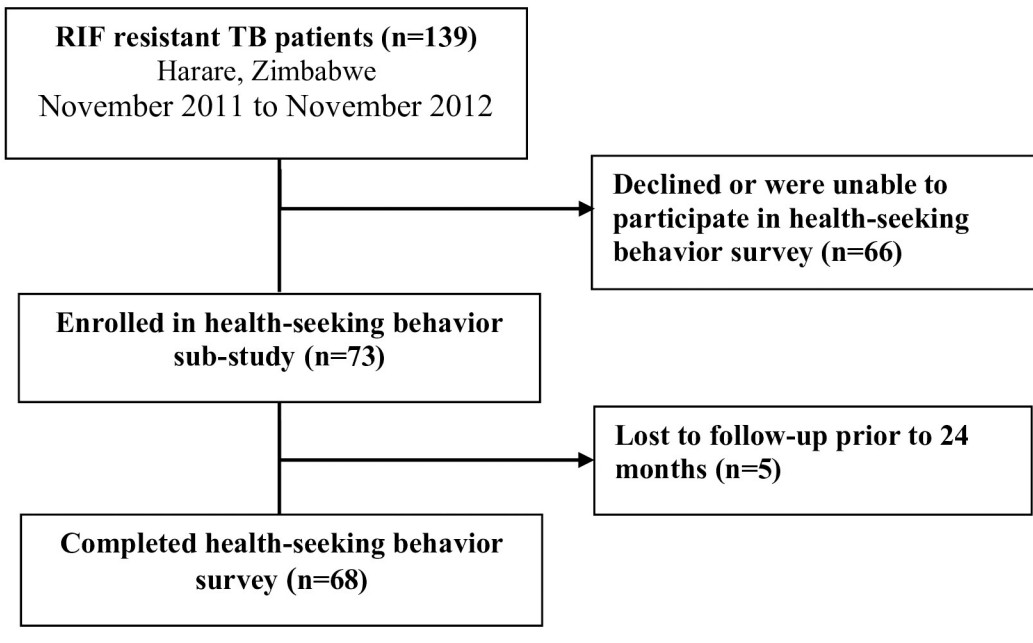

**Fig 1. Participant flowchart (n = 68).**

times; inadequate testing facilities or poor service at the clinic as reasons for not first seeking care at the clinic.

## Pathways to care and health-seeking visits

Based on qualitative interviews conducted among patients, urban residents preferred to visit pharmacies and supermarkets for home-based and over-the-counter remedies before visiting polyclinics or primary care facilities. When the care received was inadequate or symptoms were not improving, these patients would later attend government administered clinics and be referred to TB-referral clinics. Rural patients from outside Harare had a different typical care pathway characterised by home-based care, attendance at a rural hospital or clinic, followed by referral to a RR-TB treatment facility in Harare (Fig 2).

## Healthcare facilities attended

Health-seeking behaviour among study participants was cyclic, with participants making up to six visits (median, three visits) before initiating RR-TB treatment. Repeat visits at the same health facility were more common for polyclinics and private healthcare facilities. For their first visit, most participants attended a polyclinic (37%) followed by a pharmacy (30%), and private health care providers (24%) (Fig 2). Cumulatively, for the total 215 health care visits recorded in the study, nearly half (n = 90/215, 42%) were at the polyclinic with participants making up to five polyclinic visits (Fig 3).

## Association of total delay with clinical outcomes

We found that after adjusting for age and gender, living with HIV and being employed were associated with shorter total delay, (p = 0.11, median 98 days and p = 0.25, median 104 days respectively), and TB retreatment was likely to be associated with a longer delay (p = 0.20). We found no significant associations with mortality and median total delay or total cost incurred

**Table 1. Socio-demographic and clinical characteristics of RR-TB patients, Harare, Zimbabwe.**

| Variable Name | Total sample (N = 68) |
|---|---|
| Female, n (%) | 48 (70.6) |
| Age, years, median (IQR) | 34 (29–42) |
| Referred from outside Harare, n (%) | 12 (19.4) |
| Highest education*, n = 66 | |
| Primary | 11 (16.2) |
| Secondary | 50 (73.5) |
| Tertiary | 5 (7.6) |
| Occupation*, n = 67 | |
| Informal employment | 30 (44.8) |
| Formal | 7 (10.5) |
| Unemployed | 20 (29.9) |
| Student | 10 (14.9) |
| Monthly income, $US, median (IQR) | 175 (100–300) |
| HIV positive, n (%) | 43 (65.2) |
| On ART | 31 (73.8) |
| TB history | |
| New TB | 37 (55.2) |
| Retreatment TB | 30 (44.8) |
| Number of Prior TB episodes, n (%) * | N = 67 |
| 0 | 37 (55.2) |
| 1 | 20 (29.9) |
| 2 | 8 (11.9) |
| 3 | 2 (3.0) |
| TB symptoms at presentation, n (%) | |
| Cough | 68 (100) |
| Fever | 45 (66.2) |
| Weight loss | 54 (79.4) |
| Night sweats | 48 70.6) |
| | 49 |

*Missing Information: HIV status, n = 2; Education, n = 1; Occupation, n = 1; TB history, n = 1; Prior TB incidents, n = 1.

(Table 3). However, our sample size limited our ability to make conclusive determinations about the associations between delay and the risk factors that were assessed.

## Discussion

Our study demonstrates that substantial delays in both seeking and accessing care for persons with confirmed RR-TB are common. Overall delay was highest among patients first attending government polyclinics, with a median of seven months (221 days) from onset of symptoms to effective treatment. Most participants first sought care at private health care facilities outside of the national TB program, citing long waiting periods, mistrust and inadequate facilities and services as barriers to accessing public sector care. Patient costs averaged approximately 13% of monthly income with at least 30% of the households incurring costs that may be considered "catastrophic".

Our study showed community delays to effective rifampicin-resistant TB treatment despite the availability of molecular TB diagnostics such as Xpert. These findings are in agreement

**Table 2. RR-TB patients' first visit to a health care facility, patient time delays, total delay and associated costs to begin effective MDR-TB treatment.**

| Health facility visited | Patient delay days, median (IQR) | Health system delay days, median (IQR) | Total delay days, median (IQR) | First visit cost $US, median (IQR) | Total cost $US, median (IQR) |
|---|---|---|---|---|---|
| Pharmacy | 21 (14–26) | 97 (67–210) | 116 (88–247) | 10 (5–18) | 45 (19–80) |
| Government polyclinic | 30 (14–60) | 150 (18–300) | 221 (45–338) | 0 (0–4) | 8 (5–16) |
| Centralised RR-TB clinic | 18 (14–38) | 56 (33–163) | 58 (21–81) | 6 (5–13) | 26 (5–40) |
| Private clinic | 30 (18–70) | 102 (30–154) | 139 (45–226) | 30 (14–45) | 40 (15–56) |
| Other† | 30 (21–30) | 52 (9–248) | 105 (56–278) | 8 (1–14) | 14 (6–21) |
| Overall | 26 (14–42) | 97 (30–215) | 132 (51–287) | 6 (2–16) | 23 (10–55) |
| p-value | 0.084 | 0.786 | 0.586 | **<0.001** | **0.004** |

†HIV testing clinic (n = 1), herbalist (n = 1), project clinic (n = 1), grocery shop (n = 2), missing (n = 1).

with studies from this era that have documented similar delays in accessing care. Total treatment delay among TB patients were shown to be 126 days in an urban study [20]; 70 days in a rural study [21] and up to 170 days in a rural study [22] in three South African TB studies. Although our study was in an urban setting and among RR-TB patients, we observed patterns of prolonged delays averaging 97 days (overall) for health system, and as high as 150 days for polyclinics. While for the South African studies delays to treatment initiation were mainly attributed to late presentation, [21, 22] in our study, delays were largely health system related. In contrast, a systematic review on 23 studies from India reported shorter average patient delays of 18.4 days and total delays of 55 days from when symptomatic patients with active TB first contacted a health care provider [23]. Differences between the Indian studies and our findings could be attributed to the health care system setting in the different countries. Across a number of other studies, delays in treatment initiation were mainly attributed to a combination of diagnostic delays and late presentation for care [23–26]. Prolonged diagnostic and

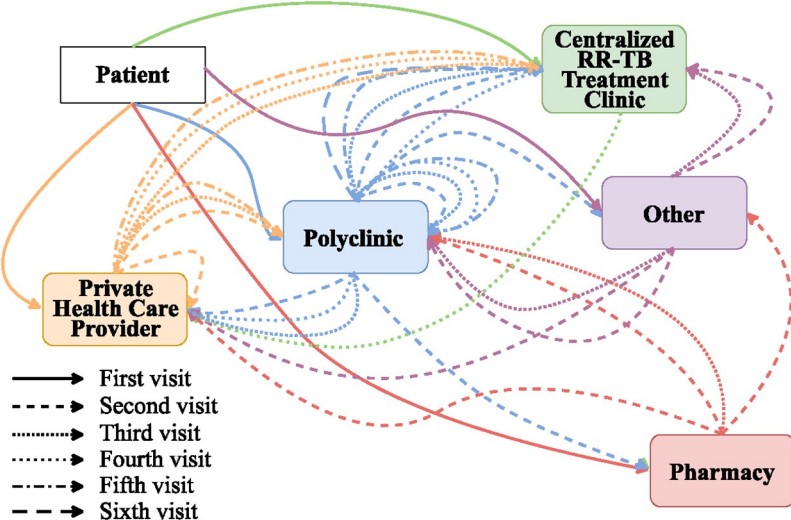

**Fig 2. Cyclic care-seeking pathways and repetitive visits followed by study participants.** Different colours correspond to the health facility where patients sought care. The different arrows represent successive care-seeking visits as shown in the figure key.

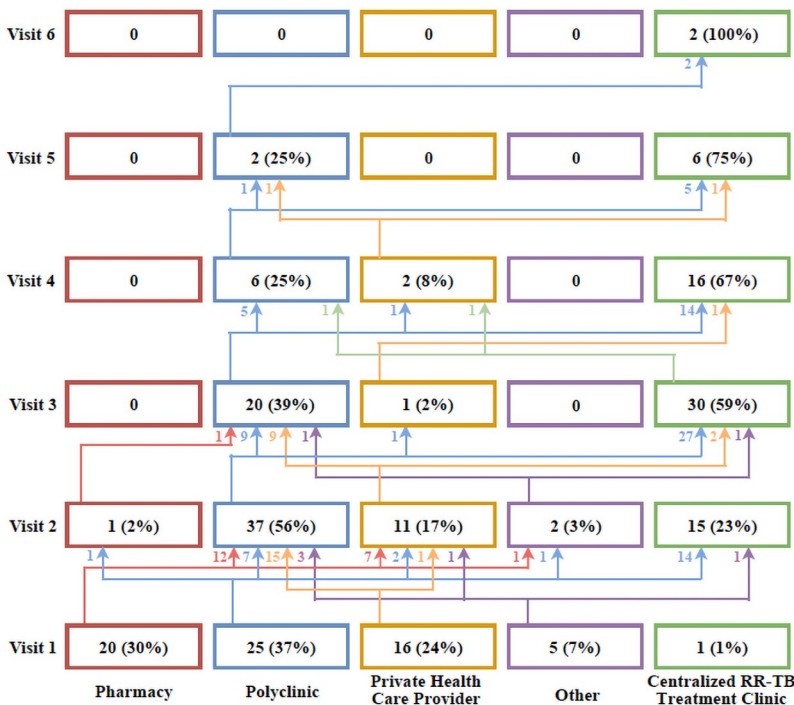

**Fig 3. Different Health care facilities attended by study participants and the proportion (%) of participants attending each facility as a proportion of all health care visits for each health care visit (Visit 1–6).** Different colours correspond to the health facility where patients sought care while different arrows represent successive care-seeking visits. *Four patients were admitted to hospital on first contact with the health system and are not included in this figure.

treatment initiation delays pose serious challenges as TB treatment may only be initiated after disease has progressed, potentially worsening prognosis and treatment outcomes [27, 28].

Indirect costs remain a deterrent to seeking health care for many patients, despite TB treatment being nominally free in government facilities in most countries [11, 29, 30]. Loss of income due to time taken when sick or to seek treatment is a major issue and an indirect expense particularly when employment is informal or seasonal. The majority of our study participants who delayed seeking treatment were informally employed with limited safety nets. It is of concern that many patients visited private health care facilities or pharmacies as their first point contact, incurring additional expenses in the process. Our findings corroborate studies in other settings where patients incurred higher costs and a circuitous path to care, [24] resulting in significant delays in accessing treatment [31]. The economic burden of TB treatment remains considerable, especially in poorer and developing countries without universal health coverage (UHC), including our study setting [5, 32]. For many, the cost of accessing care can

**Table 3. Association between delay, cost and 24-month survival.**

|  | Died (n = 6) | Survived (n = 62) | p-value* |
|---|---|---|---|
| Total delay (days) | 168.5 (83.3–309.25) | 132.0 (49.8–280.8) | 0.77 |
| Patient delay (days) | 30 (23.3–30) | 21 (14–42) | 0.62 |
| Total cost (dollars, US) | 24.5 (8–40.3) | 21.5 (10.3–55.5) | 0.75 |

*Mann-Whitney test.

push already poor families to the brink of poverty. Furthermore, as we have shown, repeat visits in care-seeking pathways increased cumulative costs and delayed diagnosis and treatment initiation. Studies from India, Indonesia, Thailand, South Africa and other similar settings have shown that the cost of TB can be catastrophic especially for low-income households [6, 12, 32–35]. According to WHO, catastrophic costs are incurred when patients spend a considerable proportion of their annual income on both direct and indirect medical costs [34]. Other unaccounted (indirect) costs for TB may include income loss due to seeking treatment and loss of productivity due to illness or when household members assume carer roles for a bedridden TB patient. The financial burden of TB is evident from the low average income per household and proportion of costs (up to a maximum of 31% of annual household income) incurred on TB-treatment costs reported by participants in this study. Up to 30% of households reportedly spent more than 20% of annual household income on TB related costs, a threshold which has been previously described as catastrophic [12].

In 2016, the WHO issued revised guidance recommending MDR-TB treatment for all RR-TB patients regardless of confirmatory resistance testing as well as the use of line probe assays for drug sensitivity testing [36]. At the time of our study, MDR-TB diagnosis was done by culture, and drug susceptibility testing was done to confirm isoniazid resistance at a central reference laboratory (although many patients would be treated on the basis of rifampicin resistant test), increasing delays to treatment initiation. Our findings point to a need to improve public sector care services to ensure that they are the preferred first contact for TB patients and that patients access treatment promptly once they enter the care cascade. This is particularly pertinent given the protracted delay in initiating treatment by patients attending polyclinics that we report in this study. Increased case-finding and educational campaigns would assist in reducing patient-associated delays in seeking care. Furthermore, supportive social protection policies and interventions such as UHC [37] in resource-constrained settings could ensure that TB patients are not deterred from seeking care by exorbitant out-of-pocket costs which may exert financial burden on impoverished families [12, 33]. Our findings corroborate previous studies from other TB burdened settings that showed that consulting private health care providers aggravate delays in TB treatment access as they are not equipped to promptly diagnose and initiate TB treatment [24, 26, 38, 39]. Private-public health partnerships could facilitate prompt TB diagnosis, reduce high costs and reduce treatment delays among presumptive MDR-TB and TB patients as has been previously reported in other studies [40–42]. Taken together, these findings point to a need for interventions including quality improvement (particularly in the public healthcare system), to reduce delays in MDR-TB and RR-TB patient diagnosis and ensure prompt treatment once patients make contact with a health care facility.

We acknowledge a few limitations in our study. First, we conducted this study within a prospective cohort of RR-TB patients who underwent extensive diagnostic testing at time of referral for RR-T testing and treatment; true programmatic delays may be longer. Second, our study was underpowered to determine associations between clinical outcomes and health-seeking behaviour-related delays. Third, female participants were over-represented in our study, [17] indicating a possible selection bias. Furthermore, data was self-reported and so may be subject to self-report bias.

In conclusion, we found significant delays in accessing TB care in this setting, particularly in the public health sector. The majority of patients preferred private care providers as their first contact, resulting in cyclical care-seeking pathways, significant delays and additional health-care costs, which did not correlate with prompt access to care. As pharmacies are the first point of contact for many patients, we recommend that pharmacists and other private health care providers should be trained in appropriate TB screening and referral of potential

TB patients to hasten access to treatment and reduce the cyclic pathways evident in settings such as ours.

## Supporting information

**S1 Data.**
(XLSX)

## Acknowledgments

We thank the Harare City Health Department clinicians and staff, who have shown exceptional dedication in treating MDR-TB and RR-TB patients despite resource constraints. We would like to acknowledge all study participants and their families. We thank Morna Cornell for critically reviewing this manuscript.

## Author Contributions

**Conceptualization:** John Z. Metcalfe.

**Data curation:** John Z. Metcalfe.

**Formal analysis:** Rebecca Tadokera, Stella Huo, Grant Theron, Collins Timire.

**Funding acquisition:** John Z. Metcalfe.

**Investigation:** Salome Manyau-Makumbirofa, John Z. Metcalfe.

**Methodology:** Salome Manyau-Makumbirofa.

**Resources:** John Z. Metcalfe.

**Supervision:** John Z. Metcalfe.

**Visualization:** Stella Huo.

**Writing – original draft:** Rebecca Tadokera.

**Writing – review & editing:** Rebecca Tadokera, Stella Huo, Grant Theron, Collins Timire, Salome Manyau-Makumbirofa, John Z. Metcalfe.

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
