## [Decision Letter · Decision Letter 0]

27 Nov 2020

PONE-D-20-13834

Health care seeking patterns among patients investigated for drug resistant tuberculosis in Harare, Zimbabwe: a prospective cohort study.

PLOS ONE

Dear Dr. Tadokera,

Thank you for submitting your manuscript to PLOS ONE. After careful consideration, we feel that it has merit but does not fully meet PLOS ONE’s publication criteria as it currently stands. Therefore, we invite you to submit a revised version of the manuscript that addresses the points raised during the review process.

We look forward to receiving your revised manuscript.

Kind regards,

Nicky McCreesh

Academic Editor

PLOS ONE

Journal Requirements:

2. Please address the following:

- Please include additional information regarding the survey or questionnaire used in the study and ensure that you have provided sufficient details that others could replicate the analyses. For instance, if you developed a questionnaire as part of this study and it is not under a copyright more restrictive than CC-BY, please include a copy, in both the original language and English, as Supporting Information. In addition, please include further details of the development and validation of this tool.

- Please refer to any sample size calculations performed prior to participant recruitment. If these were not performed please justify the reasons. Please refer to our statistical reporting guidelines for assistance (https://journals.plos.org/plosone/s/submission-guidelines.#loc-statistical-reporting).

3.We note that you have indicated that data from this study are available upon request. PLOS only allows data to be available upon request if there are legal or ethical restrictions on sharing data publicly. For information on unacceptable data access restrictions, please see http://journals.plos.org/plosone/s/data-availability#loc-unacceptable-data-access-restrictions.

Reviewers' comments:

Reviewer's Responses to Questions

**Comments to the Author**

1. Is the manuscript technically sound, and do the data support the conclusions?

Reviewer #1: Partly

Reviewer #2: Partly

2. Has the statistical analysis been performed appropriately and rigorously? 

Reviewer #1: Yes

Reviewer #2: I Don't Know

3. Have the authors made all data underlying the findings in their manuscript fully available?

Reviewer #1: Yes

Reviewer #2: No

4. Is the manuscript presented in an intelligible fashion and written in standard English?

Reviewer #1: Yes

Reviewer #2: Yes

5. Review Comments to the Author

Reviewer #1: Abstract

Pg. 2, L25: are major impediments……

Pg. 2, L28: using s prospective cohort study design

More information on data collection instruments, analysis etc. are required in the methods section

Pg. 2, L33-34 should come first in the presentation of results.

Pg. 2, L29: The socio-demographic characteristics of the participants is missing. It is important for the subject under study

Introduction

Pg. 3, L50-53: Consider making 2 sentences for better coherence

Pg. 3, L53: Remove ‘due to’

Pg. 3, L54: add ‘also’ after may

Pg. 3, L56: Change from…. in the community, which may TO …. in the community and this could….. MAY has been used severally in that paragraph

Pg. 3, L59: add the before availability

Pg. 3, L71: Study area/setting missing

Pg. 4, L87-92: Consider breaking long sentences

Pg. 4, L87, 96: Source of the questionnaire and definitions lacking. More information needed

Results

Pg. 5, L122: More information on this main study is needed in the preceding sections

Table 1: Many variable son this table don’t add up to 68: e.g. age categories, occupation etc.

Pg. 9, L161; This information needs to be reflected in the abstract

Pg. 9, L164; Participants who did not first care at…..

Pg. 9, L168: Qualitative data collection and analysis was not described in the methods, how come these results?

Pg. 9, L176: Figure legend present but figure absent

How does the information presented in L143 differ from L184

Pg. 9, L183: Remove space between visits and recorded

Pg.9, L186: Figure legend present but figure absent

Pg.9, L190: This should have been defined under methods. Pls add clinical outcomes to the definitions. How many people died or survived?

Pg. 11, L212: contacted

Pg.11, L233, 236 : Integrate ref 5 and 29 as one statement as both sentences refer to the same thing

Pg. 12, L241: for a bedridden….

Pg. 12, L254: or by exorbitant….

L257: another recommendation could be integrating private providers in care provision given their high patronage

L264; use because instead of to the extent that

L271, 272: Interviews not discussed in methods section

L275: self-report is another limitation

L281: not discussed in the discussion, see comment above

L284: We recommend the improvement of the quality of care………in order to reduce……….and ensure more prompt……………

Reviewer #2: PONE-D-20-13834

Health care seeking patterns among patients investigated for drug resistant tuberculosis in Harare, Zimbabwe: a prospective cohort study

Thank you for a really interesting and important piece of work. Suggestions below are merely to help you strengthen your work and the final product. I have added comments with reference to specific sections of the work below.

Abstract:

Would be good if you could specify what your data source is; as well as outline your study design which I think is a mixed methods study (cross-sectional quantitative data) with qualitative interviews?

General comments:

Please define all abbreviations used in your manuscript.

Page 3, line 64: I think you mean disability, instead of debility.

Page 4, line 74: how many facilities were sampled and how were these facilities selected?

Patients were interviewed at selected health facilities and identified based on the selection criteria outlined.

Page 4, line 87: should read “data were collected…”

Page 4, line 91: where were the other variables collected from if not directly from the patient through patient interviews?

Page 5: under your statistical analysis paragraphs, it might be useful to specify different types of analyses separately. So, for example, perhaps start with describing how you would do your initial descriptive analysis where you are describing your sample. Specify if you are comparing different types of patients in your sample with one another. Then describe how you are setting up your regression analysis where you are wanting to explain your findings related to the time spent in accessing care or you are wanting to understand which variables and how they explain patient and health seeking delays. Then explain which statistical approaches you used and why.

Page 5, line 106: Why did you categorise your continuous variables, was that for the initial reporting in Table 1? In your statistical analysis, it is better if you can use the continuous variables in your model.

Page 5, line 124: It would be helpful if you could report not only the percentage, but to also show the numerator and denominator for the percentage reported, ex. 45% (31 /68) of participants had a history of TB.

On page 6, line 126: How do you define informal employment. It is worth spelling this out a bit more for the reader.

Page 6, line 128: Figure 1 is not showing up in the review document. What were the reasons for patients to not consent to the health seeking behaviour survey? How did your sample reduce from 73 following initial consent to 68 included in your analysis?

Table 1:

High proportion of the sample was female – why is this? Is it a possible sampling bias?

What is the relevance of having been referred from outside Harare? A more detailed description of the setting and DR-TB services in Harare will be needed in your background so that this statement makes sense. Plus, you need to remember that your readers may not know what the health system in Zimbabwe is like, and how people move through care. How do they pay for care? In your introduction, it would strengthen your paper if you could share some of your knowledge of how the system works to give the reader a sense of why your research question is important and what the context is within which they need to understand the results.

When you describe the delays; it would be good to first describe patient delay (as you have done), then health facility delay and then the total delay. This allows the reader to follow how you got to your total delay estimate.

Do you perhaps have any data on what proportion of people first started drug-sensitive TB treatment (DS-TB) before initiating on rifampicin resistant TB treatment?

Page 9, line 161: the results here sounds as if they came from open-ended questions/ or from qualitative work. If so, you need to be clear with the reader where that data came from and how it was analysed. This needs a separate section in the methods section with detailed discussion and justification of the methods used.

Page 9, line 174: Figures 2 and 3 are missing.

In the section “Associations with clinical outcomes” you are reporting important and interesting results. But I think this part of your analysis needs to be more clearly described in your methods section – specifically here I am referring to your comparison of costs and health seeking delays between who die and those who survive. How was this outcome (death) collected and what are the limitations of the approach used? Next, how did you set up your regression analysis and then in the results even if those results are not significant it would still be useful to know what the direction of the relationship between those variables is and even if not significant you can still report the p-value. It would strengthen your paper if you could explore what the determinants of long delays to RR-TB treatment are… for example, it might be accessing a certain facility first, or age or gender… But if you set up your regression analysis to answer this question, the answers would be very helpful to policy makers as it would allow them to construct useful and data-informed interventions.

Your discussion section is great – the work is well positioned. The paper would be even stronger if you added a sentence or two on what your think the policy implications of the work are. What is your advice to the NTP, how should they use your results to improve the service?

6. PLOS authors have the option to publish the peer review history of their article (what does this mean?). If published, this will include your full peer review and any attached files.

Reviewer #1: No

Reviewer #2: No

---

## [Author Response · Author response to Decision Letter 0]

4 Mar 2021

We have addressed all comments as requested by the Editor and the reviewers (see details in responses to reviewers). Furthermore, we have included a copy of our anonymised data as well as the study questionnaires as requested with submission pack.

---

## [Decision Letter · Decision Letter 1]

13 Apr 2021

PONE-D-20-13834R1

Health care seeking patterns of rifampicin-resistant tuberculosis patients in Harare, Zimbabwe: a prospective cohort study.

PLOS ONE

Dear Dr. Tadokera,

Thank you for submitting your manuscript to PLOS ONE. After careful consideration, we feel that it has merit but does not fully meet PLOS ONE’s publication criteria as it currently stands. Therefore, we invite you to submit a revised version of the manuscript that addresses the points raised during the review process.

We look forward to receiving your revised manuscript.

Kind regards,

Anna Maria Mandalakas, MD

Academic Editor

PLOS ONE

Journal Requirements:

Reviewers' comments:

Reviewer's Responses to Questions

**Comments to the Author**

1. If the authors have adequately addressed your comments raised in a previous round of review and you feel that this manuscript is now acceptable for publication, you may indicate that here to bypass the “Comments to the Author” section, enter your conflict of interest statement in the “Confidential to Editor” section, and submit your "Accept" recommendation.

Reviewer #2: (No Response)

2. Is the manuscript technically sound, and do the data support the conclusions?

Reviewer #2: Yes

3. Has the statistical analysis been performed appropriately and rigorously? 

Reviewer #2: Yes

4. Have the authors made all data underlying the findings in their manuscript fully available?

Reviewer #2: No

5. Is the manuscript presented in an intelligible fashion and written in standard English?

Reviewer #2: Yes

6. Review Comments to the Author

Reviewer #2: PONE-D-20-13834R1

My comments were sufficiently addressed. Additional comments, with further information:

In Table 2: show days per unit of time.

Patient population: Rif-resistant

Page 2, from line 35: better to quote the arithmetic mean of costs (in addition to the median). Cost data are typically right skewed. The arithmetic mean is used when we estimate the total cost to a budget (mean cost x number of patients to be treated). This is a common use of health service cost data when considering policy options.

A paper of interest is Thompson & Barber. 2000. How should cost data in pragmatic randomised trials be analysed? https://www.ncbi.nlm.nih.gov/pmc/articles/PMC1127588/.

Page 3, line 72: prospective cohort study nested in what?

Page 9, line 202: would be good to specify the variables you adjusted for

Page 9, line217 and page 8, line 173: costs as a percentage of annual income is shown. It would be helpful if you could also report the proportion of patients who faced catastrophic costs. This will bring your paper in line with current literature. For an overview of the topic, see http://www.stoptb.org/wg/dots_expansion/tbandpoverty/assets/documents/Tool%20to%20estimate%20Patients%27%20Costs.pdf.

7. PLOS authors have the option to publish the peer review history of their article (what does this mean?). If published, this will include your full peer review and any attached files.

Reviewer #2: No

---

## [Author Response · Author response to Decision Letter 1]

17 May 2021

4. Have the authors made all data underlying the findings in their manuscript fully available?

Reviewer #2: No

We have submitted our dataset as per PLOS Data policy and as requested in the previous review (File named: Anonymized Study Data). 

In Table 2: show days per unit of time. 

We have corrected Table 2 and added the units (days) as suggested by the reviewer.

Patient population: Rif-resistant.

We have also corrected the table heading to include the patient population description, “RR-TB Patients’’. (page 21)

Page 2, from line 35: better to quote the arithmetic mean of costs (in addition to the median). 

We have amended this sentence to read “at a median cost of 13% (IQR 6-31%) of their total annual household income (mean cost, US$410” (line 32-33). 

Likewise, we have also amended line 172-173 to read “Overall, a median of 13% (IQR 6-31%) or mean cost of US$410 of total annual household income”.

Page 3, line 72: prospective cohort study nested in what?

We have amended this sentence to simply read “We conducted a prospective cohort study….” (line 72)

Page 9, line 202: would be good to specify the variables you adjusted for

We have amended this statement to read “We found that after adjusting for age and gender....” (line 203)

---

## [Editor Report · Decision Letter 2]

26 May 2021

PONE-D-20-13834R2

Health care seeking patterns of rifampicin-resistant tuberculosis patients in Harare, Zimbabwe: a prospective cohort study.

PLOS ONE

Dear Dr. Tadokera,

Thank you for submitting your manuscript to PLOS ONE. After careful consideration, we feel that it has merit but does not fully meet PLOS ONE’s publication criteria as it currently stands. Therefore, we invite you to submit a revised version of the manuscript that addresses the points raised during the review process.

We look forward to receiving your revised manuscript.

Kind regards,

Anna Maria Mandalakas, MD

Academic Editor

PLOS ONE

Journal Requirements:

Additional Editor Comments (if provided):

The team has done an excellent job with revisions. The manuscript looks quite good; I am excited to see this published.

I have one small suggestion for your consideration:

Line 304 (concluding paragraph): please consider revising by inserting the word 'screening' such that the revised text states "should be trained in appropriate TB screening and diagnosis, and referral of potential TB patients...."

The addition of the addition of 'screening' is perhaps more appropriate for the pharmacy level of care.

---

## [Author Response · Author response to Decision Letter 2]

2 Jun 2021

We have made the necessary revision as suggested by the Editor. We look forward to seeing our manuscript published in your Journal.

---

## [Editor Report · Decision Letter 3]

23 Jun 2021

Health care seeking patterns of rifampicin-resistant tuberculosis patients in Harare, Zimbabwe: a prospective cohort study.

PONE-D-20-13834R3

Dear Dr. Tadokera,

We’re pleased to inform you that your manuscript has been judged scientifically suitable for publication and will be formally accepted for publication once it meets all outstanding technical requirements.

Kind regards,

Anna Maria Mandalakas, MD

Guest Editor

PLOS ONE
---

## [Editor Report · Acceptance letter]

8 Jul 2021

PONE-D-20-13834R3 

Health care seeking patterns of rifampicin-resistant tuberculosis patients in Harare, Zimbabwe: a prospective cohort study. 

Dear Dr. Tadokera:

I'm pleased to inform you that your manuscript has been deemed suitable for publication in PLOS ONE. Congratulations! Your manuscript is now with our production department. 

Kind regards, 

on behalf of

Professor Anna Maria Mandalakas 

Guest Editor

PLOS ONE